# Determinants of domestic violence against women in Cambodia: How digital access, media exposure, motorcycle ownership, and partners' alcohol use matter

**Samnang Um[1]\*, Sopheap Suong[2], Chantrea Sieng[3], Sovandara Heng[4], Grace Marie Ku[5,6], Sothy Heng[7]**

**1** National Institute of Public Health, Phnom Penh, Cambodia, **2** Flinders University, Adelaide, South Australia, Australia, **3** Calmette Hospital, Phnom Penh, Cambodia, **4** Department of Mental Health and Substance Abuse, Khmer Soviet Friendship Hospital, Phnom Penh, Cambodia, **5** Department of Public Health, Institute of Tropical Medicine, Antwerp, Belgium, **6** Faculty of Medicine & Pharmacy, Vrije Universiteit Brussel, Brussels, Belgium, **7** Ministry of Health, Phnom Penh, Cambodia

\* umsamnang56@gmail.com

## Abstract

Domestic violence against women remains a public health and socio-economic burden in Cambodia, with only slow declines over the past two decades. This study examined how digital access, media exposure, motorcycle ownership, and partners' alcohol use are associated with intimate partner violence (IPV), defined as the experience of any sexual, physical, or emotional violence by a current or former partner within the past 12 months, adjusted for socio-demographic factors. A cross-sectional analysis of 5,780 weighted women aged 15–49 from the 2021–2022 Cambodia Demographic and Health Survey. IPV was regressed on mobile phone ownership, internet use, media exposure, motorcycle ownership, and partners' alcohol use using survey-adjusted multivariable logistic models. Overall, 13.2% of women reported experiencing IPV in the past year, specifically emotional violence (12.2%), physical violence (4.4%), and sexual violence (1.9%). Smartphone ownership was associated with lower odds of emotional violence (AOR = 0.7; 95% CI 0.5–0.9) and IPV (AOR = 0.7; 95% CI: 0.5–1.0), whereas low-frequency internet use predicted higher odds of emotional violence (AOR = 1.7; 95% CI: 1.1–2.7) and IPV (AOR = 1.6; 95% CI: 1.1–2.5). Partner alcohol use was a strong risk factor for IPV (AOR = 3.0; 95% CI: 2.1–4.1 and all forms: sexual (AOR = 3.5; 95% CI: 1.1–11.4), physical (AOR = 5.6; 95% CI: 2.8–11.5), and emotional (AOR = 3.1; 95% CI: 2.2–4.4). Women in wealthier households had significantly lower odds of IPV (AOR = 0.6; 95% CI: 0.5–0.8), and specifically, physical violence (AOR = 0.4; 95% CI: 0.3–0.7). These findings highlight the dual role of digital inclusion—smartphones may enhance women's protection, while limited or monitored internet access could heighten risk. Policies should be formulated to prioritize safe and private digital access, integrate gender-sensitive digital

**Data availability statement:** The data used in this study are publicly available from The DHS Program. The Cambodia Demographic and Health Survey (CDHS) 2021–2022 dataset can be accessed upon request from the DHS Program website (https://www.dhsprogram.com/data/available-datasets.cfm). Researchers must register and request access to the dataset. The authors did not receive any special privileges in accessing the data.

**Funding:** The author(s) received no specific funding for this work.

**Competing interests:** The authors have declared that no competing interests exist.

literacy, and strengthen alcohol control, and women's economic empowerment within Cambodia's National Action Plan to Prevent Violence Against Women 2019–2023 and forthcoming 2024–2030 framework.

## Introduction

Domestic violence against women, particularly intimate partner violence (IPV), remains a serious public health and women's human rights concern [1]. Globally, one in three women have experienced either intimate partner physical or sexual violence in their lifetime [1]. This issue is particularly high in low- and middle-income countries, where gender inequality, economic dependency, and limited access to information increase vulnerability [2–4]. In Southeast Asia, the estimated rate of IPV stands at approximately 33% of women [1]. In Cambodia, despite policy and legal reforms, domestic violence remains a pressing issue [5,6].

The 2021–2022 Cambodia Demographic and Health Survey (CDHS) showed that 22% of Cambodian women aged 15–49 had experienced with intimate partner violence (IPV), encompassing three main forms—physical, sexual, and emotional—committed by a current or former partner within the 12 months before the survey [7]. More specifically, 17% of women reported physical violence, 15% emotional violence, and 4% sexual violence from their intimate partners [7]. These forms of violence often overlap, and many women endure more than one type of violence [1–4,6–8]. The long-term impacts are profound. Beyond physical injuries, survivors face higher risks of depression, anxiety, reproductive health problems, and social isolation. Domestic violence also restricts women's economic participation and reinforces cycles of poverty and dependence. In Cambodia, data suggest a slow decline in IPV prevalence. For instance, physical violence reported by ever-partnered women dropped from 23% in 2005 to 20% in 2014 and 17% in 2021–22, indicating gradual progress [7–9]. The persistence of such violence—especially in rural areas—signals the need for more targeted, structural interventions [7–9].

The Royal Government of Cambodia has prioritized the elimination of violence against women through the previous and current National Action Plan to Prevent Violence Against Women, 2019–2023, published on 9 October 2020. The Plan identifies four strategic areas—prevention, legal protection and multi-sectoral services, formulation and implementation of laws and policies, and review, monitoring, and evaluation—to strengthen coordinated national action against gender-based violence [10]. Recognizing that gender inequality underlies violence, it also highlights digital technology, women's empowerment, and economic participation as transformative drivers of prevention. In alignment with this framework, the Neary Rattanak VI Strategic Plan 2024–2028, launched on 8 April 2024, further reinforces national commitments to promote gender equality and empower women and girls in Cambodia [11]. This study provides empirical evidence to inform the design of the forthcoming National Action Plan to Prevent Violence Against Women 2024–2030.

This study is guided by the Ecological Model of Violence Prevention, which conceptualizes IPV as the outcome of interacting factors at multiple levels: individual (e.g., demographic characteristics, education, and alcohol use), relational (e.g., partner characteristics, family dynamics), community (e.g., social networks, resource availability), and societal (e.g., gender norms, economic, and policy structures) [12,13]. This framework allows a multidimensional analysis of IPV by integrating both risk and protective factors. At the individual level, women's access to informational and economic resources—such as mobile phones, internet, media, and transport—can enhance autonomy and help-seeking capacity, whereas partner alcohol consumption represents a relational factor that can intensify conflict and aggression [8,9,14]. Community-level and societal-level determinants, such as wealth inequality, unavailability of community support, and gender norms, further condition women's exposure to risk [2,8,9,15]. Therefore, this framework was used to examine digital access, mobility, alcohol use, socio-economic status, and rural/urban residence simultaneously to understand multi-level influences on IPV risk, and its specific violence forms.

Globally, research has consistently identified low levels of education, poverty, rural residence, and partner alcohol use as significant predictors of IPV [16–18]. Alcohol use by male partners has been one of the most robust risk factors across settings, directly linked to both the frequency and severity of abuse [16,19]. Studies in South and Southeast Asia—including Bangladesh, India, and the Philippines—have similarly shown that alcohol consumption among men, economic hardship, and patriarchal attitudes contribute to high IPV prevalence [15,20,21].

While most research on domestic violence in Cambodia have focused on socio-demographic factors such as age, education level, income, occupation, or alcohol use by partners, emerging evidence points to new areas of influence [9]. Access to digital technology, exposure to mass media, and ownership of transportation such as motorcycles are particularly being explored as potential factors that could reduce the risk of violence.

Digital access—including mobile phone and internet use—can increase women's autonomy, improve access to information and services, and provide discreet channels for help-seeking. Studies in India and Africa show a positive link between mobile phone ownership and reduced IPV risk, likely due to increased social connectedness and reduced isolation [22–26]. Similarly, exposure to mass media through radio, television, or newspapers can positively influence social norms by challenging discriminatory gender stereotypes and encouraging non-violent conflict resolution [27]. Evidence from Bangladesh and Nigeria have shown that women with regular media exposure are less tolerant of domestic violence and more likely to seek help [28–30].

Mobility, particularly through ownership of transportation such as motorcycles, may also influence women's vulnerability to IPV. In areas with poor transportation infrastructure, owning a motorcycle can enhance a woman's ability to access services, participate in the workforce, or leave abusive situations. Evidence from rural India and Southeast Asia suggests that women with access to transportation are more empowered and better positioned to seek support or exit violent relationships [31–33].

There is growing recognition of the above factors globally. Locally, Cambodia's National Action Plan to Prevent Violence Against Women 2019–2023 underscores the importance of multisectoral strategies to prevent and respond to violence against women, with a specific focus on empowerment through access to information, services, and technology [10] The plan recognizes the transformative role of digital connectivity, media, and mobility in reshaping gender norms and increasing women's agency. This is consistent with global evidence suggesting that digital inclusion, media literacy, and personal mobility may influence women's exposure to IPV. For example, mobile phone and internet access can enhance women's autonomy, improve help-seeking, and reduce isolation. Likewise, media exposure can challenge harmful social norms, and access to transportation such as motorcycles may increase women's capacity to access services or leave abusive environments. However, there is limited empirical research in Cambodia examining how digital access, media exposure, and mobility empower women and how these interact with traditional risk factors such as partner alcohol use and household wealth to shape women's vulnerability to IPV [2,24,26,27,34,35]. Crucially, the multi-sectoral analysis of these factors using the most recent national data remains a significant gap.

This study aims to address this gap by simultaneously examining the dual role of digital access, media exposure, motorcycle ownership (as a novel physical asset), and partners' alcohol use on intimate partner violence (IPV)—including physical, sexual, and emotional forms—among women in Cambodia, adjusted for sociodemographic factors. The novelty of this work, which provides a critical, contemporary evidence base, lies in its integrated focus on these emergent and protective factors alongside established risk factors within a single, rigorous framework, directly informing the implementation and future directions of national strategies like the National Action Plan to Prevent Violence Against Women [10], particularly the forthcoming 2024–2030 version, and other gender-responsive development efforts [1,10].

## Methods

### Ethical statement

The Cambodia National Ethics Committee for Human Health Research (NECHR) approved the data collection tools and procedures for CDHS 2021–2022 for Health Research on May 10, 2021 (Ref # 83 NECHR), and ICF's Institutional Review Board (IRB) in Rockville, Maryland, USA. Written informed consent was obtained from all participants prior to data collection. For respondents under 18 years of age, consent was obtained from a parent or guardian. This study used de-identified secondary data and was therefore exempt from additional institutional ethical approval.

### Data source

This study used available data from the 2021–2022 Cambodia Demographic and Health Survey (CDHS), a nationally representative cross-sectional survey conducted by the National Institute of Statistics and the Ministry of Health, with technical support from ICF International. The dataset is accessible through The DHS Program (https://dhsprogram.com/data/available-datasets.cfm) upon registration and approval. Data was collected from September 15, 2021, to February 15, 2022. Two-stage stratified cluster sampling was used to collect the samples [7]. First, 709 clusters or enumeration areas (EAs) were selected and stratified by urban-rural using probability proportional to cluster size, and second, 25–30 households were selected in each EA using systematic sampling [7]. In total, 19,496 women aged 15–49 were interviewed face-to-face using a survey questionnaire that included socio-demographic characteristics, alcohol consumption, tobacco use, household assets, maternal health-related indicators, and nutritional status [7]. The domestic violence module was administered to a subsample of 6,204 eligible women (5,780 weighted), in accordance with the WHO ethical guidelines [7,36].

### Study population

The analysis included 5,780 weighted women aged 15–49 who completed the domestic violence module and had complete data on key variables of interest [7].

### Outcome variable

The primary outcome was intimate partner violence (IPV), defined as self-reported experience of at least one of the following three forms of violence perpetrated by a heterosexual husband or partner in the past 12 months: sexual violence, physical violence, and/or emotional violence [7]. Emotional violence was specifically measured by women reporting that their partner insulted, humiliated, or threatened to hurt them or someone close to them, as detailed in the CDHS survey instrument [7].

### Independent variables

**Main exposure:** Women's characteristics included mobile phone ownership (categorized as no mobile phone, non-smartphone, and smartphone), internet use frequency (no use, low frequency, and daily or almost daily use), and media

exposure measured by the number of media types accessed (none, one, or two or more of newspaper, radio, and television). Additionally, household motorcycle ownership (yes/no). **Covariates**: women's age groups (<24, 25–34, 35–49 years), marital status (not married, married, widowed/divorced), educational attainment (no education, primary, secondary or above), and employment status (employed or not employed) were considered. Partner characteristics incorporated age groups (<24, 25–34, 35–44, ≥ 45 years), educational levels (no education, primary, secondary, or above), and alcohol use (yes/no). Household wealth status was categorized into three quintiles (poor, middle, and rich) based on CDHS-reported data [7] and place of residence (urban or rural).

## Statistical analysis

The data was analyzed using STATA v18 (Stata Corp, Texas, 2023) [37]. The standard DHS sampling weight and complex survey design were accounted for using the survey package. Descriptive statistics were used to estimate key characteristics of the study population, with results presented as weighted frequencies and percentages.

A separate bivariate analysis using chi-square tests was conducted to explore associations between independent variables (mobile phone ownership, internet use frequency, media exposure, partners' alcohol use, and other covariates) with outcome variables (sexual violence, physical violence, emotional violence, and IPV). Variables that showed a significant association (p-value ≤ 0.05) in the bivariate chi-square analysis were then included in the multiple logistic regression analysis [21].

Given that the outcome variables (IPV, sexual violence, physical violence, and emotional violence) are all dichotomous (Yes/No), multiple binary logistic regression models were used to examine the odds of women had experiencing each form of violence for each independent variable, controlling for covariate variables. Results are reported as adjusted odds ratios (AOR) with 95% confidence intervals (CI).

Multicollinearity of the independent variables (mobile phone ownership, internet use frequency, media exposure, partners' alcohol use, age, wealth index, education, and place of residence) was evaluated using the variance inflation factor (VIF) for the regression coefficients [38].

The goodness-of-fit of the logistic regression models for different forms of violence types was evaluated using the F-adjusted mean residual goodness-of-fit test.

To evaluate the potential effect modification of statistically significant associations in the adjusted analysis, these were further visualized as predicted probabilities using STATA 18, employing the margins command to estimate and visualize with a marginal plot [39].

## Results

### Descriptive characteristics of the study population

This study analyzed weighted data from 5,780 Cambodian women aged 15–49 years. Most participants (73.7%) reported owning a smartphone, while 17.1% did not own any mobile phone. About half (51.5%) used the internet daily, whereas 38.9% had never used it. More than half of the women (54.4%) reported no regular exposure to newspapers, radio, or television. Motorcycle ownership was common, reported by 87.0% of respondents. Most women were married or living with a partner (87.2%), and 70.5% reported employment. Educational levels varied: 13.7% had no formal schooling, while 42.5% had reached at least secondary school. Looking at household wealth, 44.7% lived in richer households, 35.9% in poorer ones, and 19.5% fell in the middle category. Half (50.1%) of male partners had completed secondary education or higher, while 10.4% had no schooling. Alcohol use among male partners was reported by 82.4% of participants (Table 1).

### Prevalence of intimate partner violence (IPV) and its specific forms among women

Overall, 13.2% reported experiencing intimate partner violence (IPV) in the past year, defined as experiencing physical, sexual, and/or emotional violence. Since IPV represents the experience of any of the three forms, the total prevalence is

**Table 1. Descriptive Characteristics of the Study Population (N = 5,780 weighted).**

| Variables | Frequency | Percentage (%) |
|---|---|---|
| **Women's Characteristics** | | |
| Mobile Phone Ownership | | |
| No mobile | 990 | 17.1 |
| Non-smartphone | 531 | 9.2 |
| Smartphone | 4259 | 73.7 |
| Internet Use | | |
| No internet use | 2250 | 38.9 |
| Low-frequency use | 556 | 9.6 |
| Daily/Almost daily | 2974 | 51.5 |
| Media exposure (Newspaper, radio, TV) | | |
| None | 3146 | 54.4 |
| One | 1820 | 31.5 |
| Two or more | 815 | 14.1 |
| Motorcycle Ownership | | |
| No | 750 | 13.0 |
| Yes | 5031 | 87.0 |
| Women's Age | | |
| ≤24 | 922 | 15.9 |
| 25-34 | 2155 | 37.3 |
| 35-49 | 2703 | 46.8 |
| Marital Status | | |
| Not married | 267 | 4.6 |
| Married | 5042 | 87.2 |
| Widowed/Divorced | 472 | 8.2 |
| Educational | | |
| No education | 790 | 13.7 |
| Primary | 2535 | 43.9 |
| Secondary or above | 2455 | 42.5 |
| Employment Status | | |
| No | 1708 | 29.5 |
| Yes | 4072 | 70.5 |
| **Partner's Characteristics** | | |
| Partner's Age Group | | |
| ≤24 | 343 | 6.8 |
| 25-34 | 1712 | 34.0 |
| 35-44 | 2002 | 39.7 |
| 45+ | 984 | 19.5 |
| Partner's Education | | |
| No education | 526 | 10.4 |
| Primary | 1992 | 39.5 |
| Secondary or above | 2524 | 50.1 |
| Partner's Alcohol Use | | |
| No | 1018 | 17.6 |
| Yes | 4763 | 82.4 |

*(Continued)*

**Table 1.** (Continued)

| Variables | Frequency | Percentage (%) |
|---|---|---|
| **Household Characteristics** | | |
| Wealth Index | | |
| Poor | 2073 | 35.9 |
| Middle | 1125 | 19.5 |
| Rich | 2583 | 44.7 |
| Place of Residence | | |
| Urban | 2379 | 41.1 |
| Rural | 3402 | 58.9 |

**Notes:** Survey weights are applied to obtain weighted percentages.

not the sum of the individual types of violence presented in Table 2. Emotional violence emerged as the most common type, affecting 12.2% of women. A total of 4.4% reported physical violence, while sexual violence was the least reported, with 1.9% of women disclosing such experiences (Table 2).

### Factors Associated with intimate partner violence (IPV), sexual violence, physical violence, and emotional violence among Cambodian Women aged 15–49 years in Chi-Square Analysis

Table 3 illustrates that the prevalence of intimate partner violence (IPV), sexual violence, physical violence, and/or emotional violence among ever-partnered women aged 15–49 in Cambodia varied across individual, partner, household, and contextual factors.

### Intimate partner violence (IPV)

Overall, IPV prevalence (has experienced sexual, physical, and/or emotional violence or a combination or all) was significantly higher among women with no phone (19.0%), no internet access (15.8%), and those living in rural areas (15.8%) (p < 0.001 for all). IPV increased with age, from 9.2% in women under 24 to 15.8% in those aged 35–49 (p < 0.001). Educational attainment

**Table 2.** Prevalence of Intimate Partner Violence (IPV), Sexual, Physical, and Emotional among women (N = 5,780 weighted).

| Type of Violence | No. of Women | Percentage (%) |
|---|---|---|
| Intimate Partner Violence (IPV) | | |
| None | 5019 | 86.8 |
| One or more | 761 | 13.2 |
| Sexual Violence | | |
| No | 5673 | 98.1 |
| Yes | 107 | 1.9 |
| Physical Violence | | |
| No | 5525 | 95.6 |
| Yes | 255 | 4.4 |
| Emotional Violence | | |
| No | 5076 | 87.8 |
| Yes | 704 | 12.2 |

**Notes:** Survey weights are applied to obtain weighted percentages.

**Table 3. Prevalence of Intimate Partner Violence (IPV), Sexual Violence, Physical Violence, and/or Emotional Violence Among Women by Background Characteristics using Chi-Square test, Cambodia DHS 2021–2022, (N = 5,780 weighted).**

| Variables | IPV (N = 761) | | P | Sexual (N = 107) | | P | Physical (N = 225) | | P | Emotional (N = 704) | | P |
|---|---|---|---|---|---|---|---|---|---|---|---|---|
| | Freq. | % | | Freq. | % | | Freq. | % | | Freq. | % | |
| Women's Characteristics | | | | | | | | | | | | |
| Mobile Phone Ownership | | | | | | | | | | | | |
| No mobile | 188 | 19.0 | <0.001 | 41 | 4.1 | <0.001 | 75 | 7.6 | <0.001 | 175 | 17.7 | <0.001 |
| Non-smartphone | 95 | 17.9 | | 13 | 2.4 | | 28 | 5.3 | | 91 | 17.1 | |
| Smartphone | 478 | 11.2 | | 53 | 1.2 | | 152 | 3.6 | | 438 | 10.3 | |
| Internet Use | | | | | | | | | | | | |
| No internet use | 356 | 15.8 | <0.001 | 65 | 2.9 | <0.001 | 136 | 6.0 | <0.001 | 330 | 14.7 | <0.001 |
| Low-frequency use | 103 | 18.5 | | 7 | 1.3 | | 38 | 6.8 | | 96 | 17.3 | |
| Daily/Almost daily | 303 | 10.2 | | 35 | 1.2 | | 81 | 2.7 | | 278 | 9.3 | |
| Media exposure (Newspaper, radio, TV) | | | | | | | | | | | | |
| None | 430 | 13.7 | 0.302 | 59 | 1.9 | 0.99 | 152 | 4.8 | 0.378 | 394 | 12.5 | 0.291 |
| One | 244 | 13.4 | | 33 | 1.8 | | 73 | 4.0 | | 230 | 12.6 | |
| Two or more | 88 | 10.8 | | 15 | 1.8 | | 30 | 3.7 | | 80 | 9.8 | |
| Motorcycle Ownership | | | | | | | | | | | | |
| No | 116 | 15.5 | 0.083 | 21 | 2.8 | 0.082 | 43 | 5.7 | 0.109 | 108 | 14.4 | 0.087 |
| Yes | 645 | 12.8 | | 86 | 1.7 | | 212 | 4.2 | | 597 | 11.9 | |
| Women's Age | | | | | | | | | | | | |
| <24 | 85 | 9.2 | <0.001 | 7 | 0.8 | 0.004 | 24 | 2.6 | 0.007 | 76 | 8.2 | <0.001 |
| 25-34 | 248 | 11.5 | | 34 | 1.6 | | 81 | 3.8 | | 226 | 10.5 | |
| 35-49 | 428 | 15.8 | | 67 | 2.5 | | 150 | 5.5 | | 403 | 14.9 | |
| Marital Status | | | | | | | | | | | | |
| Not married | 4 | 1.5 | <0.001 | 0 | 0.0 | 0.19 | 1 | 0.4 | 0.004 | 4 | 1.5 | <0.001 |
| Married | 690 | 13.7 | | 95 | 1.9 | | 220 | 4.4 | | 640 | 12.7 | |
| Widowed/Divorced | 67 | 14.2 | | 13 | 2.8 | | 34 | 7.2 | | 60 | 12.7 | |
| Educational | | | | | | | | | | | | |
| No education | 156 | 19.7 | <0.001 | 30 | 3.8 | 0.002 | 62 | 7.8 | <0.001 | 151 | 19.1 | <0.001 |
| Primary | 392 | 15.5 | | 47 | 1.9 | | 139 | 5.5 | | 362 | 14.3 | |
| Secondary or above | 214 | 8.7 | | 30 | 1.2 | | 54 | 2.2 | | 192 | 7.8 | |
| Employment Status | | | | | | | | | | | | |
| No | 245 | 14.3 | 0.153 | 32 | 1.9 | 0.973 | 70 | 4.1 | 0.501 | 231 | 13.5 | 0.09 |
| Yes | 516 | 12.7 | | 76 | 1.9 | | 185 | 4.5 | | 473 | 11.6 | |
| **Partner's Characteristics** | | | | | | | | | | | | |
| Partner's Age | | | | | | | | | | | | |
| ≤24 | 38 | 11.1 | 0.001 | 5 | 1.5 | 0.035 | 13 | 3.8 | 0.026 | 34 | 9.9 | <0.001 |
| 25-34 | 183 | 10.7 | | 17 | 1.0 | | 55 | 3.2 | | 166 | 9.7 | |
| 35-44 | 313 | 15.6 | | 46 | 2.3 | | 93 | 4.6 | | 291 | 14.5 | |
| 45+ | 156 | 15.8 | | 27 | 2.7 | | 59 | 6.0 | | 149 | 15.1 | |
| Partner's Education | | | | | | | | | | | | |
| No education | 129 | 24.5 | <0.001 | 18 | 3.4 | 0.045 | 45 | 8.5 | <0.001 | 120 | 22.8 | <0.001 |
| Primary | 296 | 14.9 | | 41 | 2.1 | | 97 | 4.9 | | 275 | 13.8 | |
| Secondary or above | 266 | 10.5 | | 35 | 1.4 | | 79 | 3.1 | | 246 | 9.7 | |
| Partner's Alcohol Use | | | | | | | | | | | | |
| No | 59 | 5.8 | <0.001 | 7 | 0.7 | 0.023 | 15 | 1.5 | <0.001 | 54 | 5.3 | <0.001 |

*(Continued)*

**Table 3.** (Continued)

| Variables | IPV (N = 761) | | P | Sexual (N = 107) | | P | Physical (N = 225) | | P | Emotional (N = 704) | | P |
|---|---|---|---|---|---|---|---|---|---|---|---|---|
| | Freq. | % | | Freq. | % | | Freq. | % | | Freq. | % | |
| Yes | 702 | 14.7 | | 101 | 2.1 | | 240 | 5.0 | | 650 | 13.6 | |
| **Household Characteristics** | | | | | | | | | | | | |
| Wealth Index | | | | | | | | | | | | |
| Poor | 384 | 18.5 | <0.001 | 57 | 2.8 | 0.012 | 150 | 7.2 | <0.001 | 354 | 17.1 | <0.001 |
| Middle | 151 | 13.4 | | 20 | 1.8 | | 49 | 4.4 | | 134 | 11.9 | |
| Rich | 226 | 8.8 | | 30 | 1.2 | | 56 | 2.2 | | 216 | 8.4 | |
| Place of Residence | | | | | | | | | | | | |
| Urban | 225 | 9.5 | <0.001 | 35 | 1.5 | 0.212 | 69 | 2.9 | <0.001 | 211 | 8.9 | <0.001 |
| Rural | 536 | 15.8 | | 72 | 2.1 | | 186 | 5.5 | | 493 | 14.5 | |

**Notes:** Survey weights are applied to obtain weighted percentages. * IPV: Intimate Partner Violence (includes any of the three forms: sexual, physical, or emotional). **P**: P-value

showed a protective effect: women with secondary or higher education experienced lower IPV (8.7%) than those with no education (19.7%) (p<0.001). Women whose partners had no education (24.5%) or drank alcohol (14.7%) were at greater risk. IPV was also highest among women from poor households (18.5%) compared to rich households (8.8%) (p<0.001).

## Sexual violence

Sexual violence was more commonly reported by women without access to mobile phones (4.1%) or internet (3.4%), compared to those with smartphones (2.2%) or daily internet use (1.8%) (p<0.001 for both). Women with no education (5.2%) and those whose partners had no formal education (7.2%) were also more likely to report sexual violence (p<0.001). Higher prevalence was observed among married or formerly married women (3.4–3.6%) compared to unmarried women (0.4%) (p<0.001). Partner alcohol use was a significant factor: women whose partners drank alcohol reported more sexual violence (3.5%) than those whose partners did not (1.0%) (p<0.001).

## Physical violence

Physical violence was most prevalent among women aged 35–49 (10.7%) and those from poor households (11.8%) (p<0.001). Women with no education (13.6%) and those whose partners lacked formal education (18.3%) reported higher levels of physical violence (p<0.001). Living in rural areas (10.1%) and having a partner who used alcohol (10.6%) were also associated with greater physical violence (p<0.001). Conversely, lower rates were reported among women with higher education (5.3%) and those with daily internet access (4.8%).

## Emotional violence

Emotional violence was more frequent among women aged 35–49 (9.4%) and those who were formerly married (13.3%) or currently married (7.3%) (p<0.001). Lack of education in women (9.9%) and their partners (13.1%), rural residence (7.8%), and partner alcohol use (8.3%) were all linked to higher emotional violence (p<0.001). Women with smartphones (5.7%) and daily internet access (5.3%) reported less emotional violence.

## Multicollinearity

To assess multicollinearity among the independent variables included in the regression models for sexual, physical, and emotional health outcomes, the Variance Inflation Factor (VIF) was computed (S1 Table to S4 Table). Across all three

models, the VIF values remained below the conventional threshold of 5, indicating no evidence of severe multicollinearity. Specifically, the highest VIF values were observed for age (VIF = 2.04) and partner's age (VIF = 2.00), while all other variables, including internet use, mobile phone ownership, media exposure, use of motor transportation, educational attainment of women and partners, alcohol use, household wealth, and residence, exhibited VIFs ranging from 1.01 to 1.72. The mean VIF across all models was consistently 1.48, suggesting a low overall collinearity structure. These results confirm that the multivariate regression estimates are unlikely to be distorted by multicollinearity among the predictors.

### Goodness-of-fit

The goodness-of-fit of the logistic regression models for different forms of violence types was evaluated using the F-adjusted mean residual goodness-of-fit test. The model for sexual violence demonstrated a good fit (F(9,652) = 0.63, p = 0.7720), and the emotional violence model also showed an acceptable fit (F(9,652) = 1.57, p = 0.1247). However, the model for physical violence suggested a possible misfit (F(9,652) = 2.07, p = 0.0305). Finally, the IPV model indicated an acceptable fit (F(9,652) = 1.42, p = 0.1760).

### Adjusted Odds Ratio of Digital Access, Media Exposure, and Motorcycle Ownership with Sexual Violence, Physical Violence, Emotional Violence, and Intimate Partner Violence (IPV) Among Women

The adjusted odds ratios (AOR) presented in Table 4 examine the association between digital access, media exposure, motorcycle ownership, intimate partner violence (IPV), sexual violence, physical violence, and/or emotional violence among women in Cambodia after controlling for other socio-demographic factors.

Smartphone ownership is associated with reduced odds of IPV (AOR = 0.7; 95% CI (0.5–1.0) as well as emotional violence (adjusted odds ratio [AOR] = 0.7; 95% CI (0.5–0.9), when compared to women without mobile phones. In contrast, low-frequency internet use correlates with elevated odds of experiencing emotional violence (AOR = 1.7; 95% CI (1.1–2.7) and IPV (AOR = 1.6; 95% CI (1.1–2.5) relative to women who do not use the internet.

Furthermore, women's educational attainment appears to offer protection, as those with secondary or higher education exhibit lower odds of physical violence (AOR = 0.5, 95% CI (0.3–0.9) and emotional violence (AOR = 0.7, 95% CI (0.5–1.0) compared to women without formal education. The educational level of the woman's partner is also influential; women whose partners attained formal education (primary, secondary or higher) show reduced odds of emotional violence (AOR = 0.6, 95% CI (0.5–0.9); AOR = 0.6, 95% CI (0.4–0.8), respectively) and any IPV (AOR = 0.6, 95% CI (0.5–0.8) for both) compared to those whose partners have no education.

A strong positive association was observed between the partner's alcohol use and violence types: sexual (AOR = 3.5; 95% CI (1.1–11.4), physical (AOR = 5.6; 95% CI (2.8–11.5), emotional (AOR = 3.1; 95% CI (2.2–4.4), and any IPV (AOR = 3.0; 95% CI (2.1–4.1).

Household wealth was inversely associated with IPV. Women in rich households had lower odds of physical, emotional, and IPV compared to those in the poorest households. Specifically, women in middle-wealth households had significantly lower odds of physical (AOR = 0.4; 95% CI (0.3–0.7) and IPV (AOR = 0.6; 95% CI (0.5–0.8).

In this adjusted model, motorcycle ownership and media exposure did not show statistically significant associations with IPV. Similarly, the association between women's age and partner's age with IPV was not statistically significant after adjusting for other factors. Place of residence (rural vs. urban) also did not emerge as an important predictor of IPV in this analysis.

### Interaction analyses on the factors associated with violence types

Fig 1 presents the predicted probabilities of experiencing different forms of domestic violence (sexual, physical, emotional, and intimate partner violence [IPV]) as a function of the interaction between women's mobile phone ownership status, partner's alcohol consumption, and household wealth index. The data reveal a consistent pattern of elevated predicted probabilities for all violence types when partners consumed alcohol, irrespective of women's mobile phone ownership or

**Table 4. Adjusted Odds Ratios and 95% Confidence Intervals for the Association of Digital Access, Media Exposure, and Motorcycle Ownership with Intimate Partner Violence (IPV), Sexual Violence, Physical Violence, and/or Emotional Violence Among Women in Cambodia, CDHS 2021-2022.**

| Variables | IPV | Sexual | Physical | Emotional |
|---|---|---|---|---|
| | N = 5,780 | N = 5,780 | N = 5,780 | N = 5,780 |
| | AOR (95% CI) | AOR (95% CI) | AOR (95% CI) | AOR (95% CI) |
| **Women's Characteristics** | | | | |
| Mobile Phone Ownership | | | | |
| No mobile | Ref. | Ref. | Ref. | Ref. |
| Non-smartphone | 0.9 (0.6–1.3) | 0.6 (0.2–1.4) | 0.7 (0.4–1.1) | 0.9 (0.6–1.3) |
| Smartphone | 0.7(0.5–1.0)* | 0.4 (0.2–1.1) | 0.9 (0.5–1.4) | 0.7(0.5–0.9)* |
| Internet Use | | | | |
| No internet use | Ref. | Ref. | Ref. | Ref. |
| Low-frequency use | 1.6(1.1–2.5)* | 0.6 (0.2–1.9) | 1.2 (0.7–2.0) | 1.7(1.1–2.7)* |
| Daily/Almost daily | 1.2 (0.9–1.6) | 1.0 (0.4–2.3) | 0.7 (0.4–1.2) | 1.2 (0.9–1.7) |
| Media exposure (Newspaper, radio, TV) | | | | |
| None | Ref. | Ref. | Ref. | Ref. |
| One | 1.2 (0.9–1.5) | 1.1 (0.6–2.2) | 1.0 (0.7–1.4) | 1.2 (0.9–1.5) |
| ≥Two | 1.0 (0.7–1.5) | 1.1 (0.5–2.2) | 1.1 (0.6–2.0) | 1.0 (0.7–1.5) |
| Motorcycle Ownership | | | | |
| No | Ref. | Ref. | Ref. | Ref. |
| Yes | 1.1 (0.8–1.5) | 0.8 (0.4–1.7) | 1.1 (0.7–1.7) | 1.1 (0.8–1.5) |
| Women's Age | | | | |
| 15-24 | Ref. | Ref. | Ref. | Ref. |
| 25-34 | 1.1 (0.7–1.5) | 1.5 (0.5–4.6) | 1.9 (1.0–3.8) | 1.0 (0.7–1.5) |
| 35-49 | 1.2 (0.8–1.8) | 1.7 (0.6–5.5) | 2.1 (1.0–4.5) | 1.2 (0.8–1.8) |
| Woman's Education | | | | |
| No education | Ref. | Ref. | Ref. | Ref. |
| Primary | 1.0 (0.7–1.3) | 0.7 (0.3–1.4) | 0.8 (0.6–1.3) | 0.9 (0.7–1.2) |
| Secondary or above | 0.7 (0.5–1.1) | 0.7 (0.2–1.9) | 0.5(0.3–0.9)* | 0.7(0.5–1.0)* |
| Employment Status | | | | |
| No | Ref. | Ref. | Ref. | Ref. |
| Yes | 0.9 (0.8–1.2) | 1.3 (0.7–2.5) | 1.2 (0.9–1.7) | 0.9 (0.7–1.1) |
| **Partner's Characteristics** | | | | |
| Partner's Age Group | | | | |
| ≤ 24 | Ref. | Ref. | Ref. | Ref. |
| 25-34 | 0.9 (0.6–1.5) | 0.6 (0.2–2.2) | 0.6 (0.3–1.3) | 0.9 (0.6–1.5) |
| 35-44 | 1.3 (0.7–2.2) | 1.0 (0.2–4.2) | 0.7 (0.3–1.5) | 1.3 (0.7–2.3) |
| 45+ | 1.3 (0.7–2.4) | 1.2 (0.3–4.7) | 0.9 (0.3–2.3) | 1.4 (0.7–2.5) |
| Partner's Education | | | | |
| No education | Ref. | Ref. | Ref. | Ref. |
| Primary | 0.6(0.5–0.8)** | 0.8 (0.4–1.6) | 0.7 (0.5–1.1) | 0.6(0.5–0.9)** |
| Secondary or above | 0.6(0.4–0.8)*** | 0.8 (0.3–1.8) | 0.8 (0.5–1.3) | 0.6(0.4–0.8)*** |
| Partner's Alcohol Use | | | | |
| No | Ref. | Ref. | Ref. | Ref. |
| Yes | 3.0(2.1–4.1)*** | 3.5(1.1–11.4)* | 5.6(2.8–11.5)*** | 3.1(2.2–4.4)*** |

*(Continued)*

**Table 4.** (Continued)

| Variables | IPV | Sexual | Physical | Emotional |
|---|---|---|---|---|
| | N = 5,780 | N = 5,780 | N = 5,780 | N = 5,780 |
| | AOR (95% CI) | AOR (95% CI) | AOR (95% CI) | AOR (95% CI) |
| **Household Characteristics** | | | | |
| Wealth Index | | | | |
| Poor | Ref. | Ref. | Ref. | Ref. |
| Middle | 0.7(0.6–0.9)* | 0.7 (0.4–1.4) | 0.6(0.4–1.0)* | 0.7(0.6–1.0)* |
| Rich | 0.6 (0.5–0.8)*** | 0.6 (0.2–1.3) | 0.4(0.3–0.7)*** | 0.7* (0.5–0.9) |
| Place of Residence | | | | |
| Urban | Ref. | Ref. | Ref. | Ref. |
| Rural | 1.1 (0.8–1.4) | 0.7 (0.4–1.4) | 1.0 (0.7–1.5) | 1.1 (0.8–1.4) |

**Noted:** Survey weights are applied to obtain weighted percentages: * p < 0.05, ** p < 0.01, *** p < 0.001; **AOR** = adjusted odds ratio; **CI** = confidence interval; and **Ref.** = Reference.

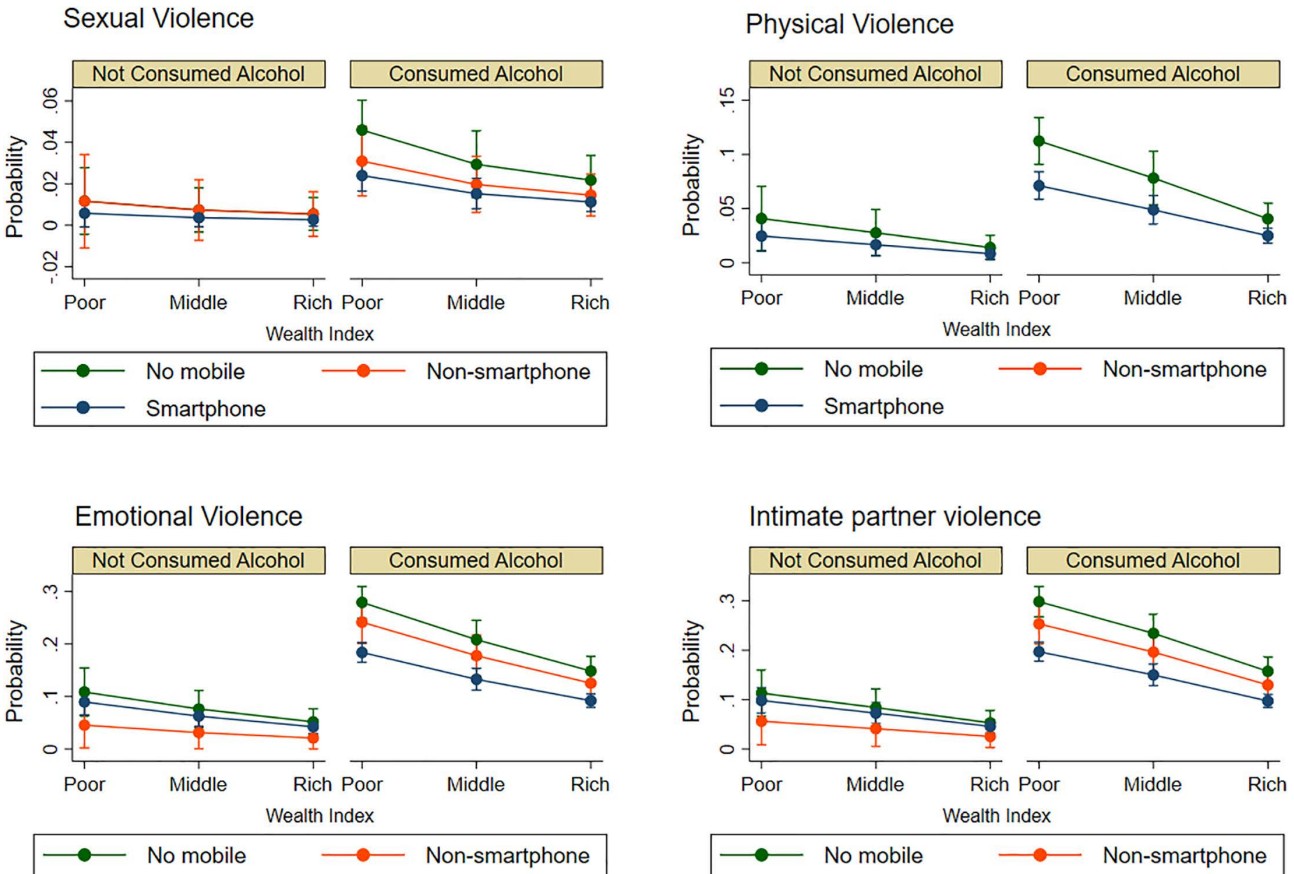

**Fig 1. Predicted probability of domestic violence types.** Illustrates the estimated probabilities resulting from the interaction between mobile phone ownership status, partner's alcohol use, and household wealth index.

household wealth index. This finding aligns with extensive literature demonstrating the robust association between partner alcohol use and increased risk of IPV in Cambodia [8,9,40]. In contrast, an inverse relationship is observed between household wealth index and the predicted probability of violence, with women in wealthier households demonstrating lower predicted probabilities across all categories of violence and partner alcohol consumption. This protective effect of economic resources has been documented in various contexts, suggesting that economic empowerment can reduce women's vulnerability to violence [14,41]. The association between mobile phone ownership and predicted probabilities is more nuanced. While women with smartphone ownership tend to exhibit lower predicted probabilities of sexual and physical violence in the absence of partner alcohol consumption, this association is less pronounced, or attenuated, when partners consume alcohol. The complex role of technology in IPV dynamics, where it can offer both protective and risk factors, is increasingly recognized [24,26,35]. These findings underscore the complex interplay of socio-economic and behavioral factors in predicting the likelihood of domestic violence.

## Discussion

This study examined the relationship between digital access, mass media exposure, motorcycle ownership, and intimate partner violence (IPV)—defined as experiencing physical, sexual, or emotional violence by a partner— among Cambodian women aged 15–49, using recent data from the 2021–22 Cambodia Demographic and Health Survey (CDHS). This study contributes novel evidence on how contemporary dimensions of women's empowerment, such as smartphone ownership, internet use, and personal mobility, interact with traditional risk factors like partner alcohol consumption and household wealth to shape IPV risk in Cambodia.

Consistent with the Ecological Model of Violence Prevention [12,13], our findings build upon the backdrop provided by recent 2021–2022 CDHS analyses, which demonstrate that IPV risk is determined by factors across multiple levels. For instance, Shaikh (2025) highlighted the geographical clustering of IPV in areas with lower female autonomy and digital connectivity, underscoring the community and societal dimensions of IPV risk [15]. Um et al. (2025) confirmed the strong association between high alcohol consumption and gender-based violence, advocating for integrated prevention programs that must address this issue at the individual (partner's use), relational, community (availability), and societal (norms/pricing) levels [40]. Similarly, Banstola et al. (2025) demonstrated the influence of media exposure on sensitive health behaviors, suggesting the potential of information channels for positive change at the societal level [27,30]. These studies identify digital access, spatial context, and substance use as critical leverage points for gender-sensitive interventions. Our analysis addresses the complex, multi-level nature of these factors, examining digital access, motorcycle ownership, and partner alcohol use simultaneously across at the individual relational.

The protective association of smartphone ownership with a 30% reduction in the odds of any IPV after adjustment aligns with patterns observed in other South Asian contexts [34]. This suggests that in Cambodia, smartphones may empower women by providing discreet access to support networks, vital information, and resources, thereby mitigating their vulnerability to violence. This acts as a protective factor at the individual level by enhancing a woman's agency and connection. Conversely, the finding that infrequent internet use (less than daily) was associated with a 60% increase in the odds of emotional and overall IPV warrants further consideration. This may be indicative of a "backlash effect," where limited and perhaps misunderstood internet engagement by women in a context of prevailing patriarchal norms can trigger suspicion, jealousy, and controlling behaviors from partners. Such intermittent use might be perceived as secretive or challenging to established traditional gender roles, leading to increased emotional abuse and overall IPV. This finding suggests that access alone is not sufficient; without corresponding digital literacy, privacy safeguards, and gender-sensitive education, connectivity can unintentionally heighten relational tension.

At the relational level, partner alcohol consumption was the strongest predictor of IPV across all outcomes (adjusted OR ≈ 4–6), consistent with prior Cambodian research [9,40]. This emphasizes the need for alcohol-harm reduction initiatives as an integral part of IPV prevention strategies.

Interestingly, mass media exposure and motorcycle ownership were not independently associated with IPV. This suggests that mere access to information or physical mobility may not overcome underlying power imbalances within households. For media exposure, the type and content of messaging likely determine its effectiveness in challenging social norms and promoting non-violent behaviors [30,33]. For motorcycle ownership, increased mobility may not reduce IPV unless accompanied by shifts in household gender dynamics and women's decision-making autonomy. These null findings highlight the importance of addressing both structural and relational determinants of violence.

At the societal and community levels, lower household wealth and education were associated with higher IPV risk, emphasizing structural inequalities as important targets for intervention. This aligns with the broader evidence that poverty alleviation, women's education, and gender-transformative policies are central to sustainable reductions in IPV [2,9,15,16].

### Policy and programmatic implications

Our findings carry important policy and programmatic implications. First, interventions should focus on expanding safe and private smartphone access while providing digital literacy and privacy training, enabling women to fully harness technology for empowerment. Second, anti-violence messaging and information on support services should be integrated into digital platforms and media channels, ensuring that content is consistent, culturally relevant, and accessible [42]. Third, alcohol-harm reduction strategies—such as community awareness campaigns, taxation measures, and counseling programs—should be linked with IPV-prevention efforts. Finally, multisectoral approaches addressing poverty and education can complement individual-level interventions, supporting the objectives of Cambodia's National Action Plan to Prevent Violence Against Women 2019–2023 and the broader Neary Rattanak VI Strategic Plan 2024–2028 [10,11]. Future research should employ longitudinal or mixed-method designs to better capture the dynamic interplay between digital empowerment, gender norms, and IPV risk in Cambodia's evolving socio-digital landscape. By situating findings within the Ecological Model, this study provides actionable evidence for integrated, multilevel interventions to reduce IPV while advancing women's autonomy and safety.

### Strengths and limitations

This study benefits from several key strengths, including the use of a large, nationally representative sample from the 2021−22 CDHS, the application of validated and standardized DHS measures for assessing IPV, and the use of survey-weighted statistical methods, including multivariable logistic regression, to account for the complex survey design and control for critical sociodemographic covariates.

Despite these strengths, several limitations warrant consideration. The cross-sectional nature of the data precludes establishing causal relationships between the examined factors and IPV. Reporting bias, inherent in studies addressing sensitive topics like IPV, may also influence the findings. Furthermore, the CDHS dataset lacks specific measures of digital autonomy, such as whether women have private access to their devices or are subject to partner control. The media exposure variable's focus on frequency rather than content represents another limitation. Finally, the study's findings are specific to ever-partnered women and may not be generalizable to unmarried or non-cohabiting individuals.

### Conclusion

This study reveals a complex relationship between digital access and IPV risk among Cambodian women. While smartphone ownership offers a protective effect, potentially by enhancing access to support and resources, infrequent internet use is associated with increased risk, possibly due to triggering partner suspicion in contexts of restrictive gender norms. The lack of a strong independent association between media exposure and motorcycle ownership underscores the importance of content and normative change alongside access. Ultimately, partner alcohol use remains a critical and potent risk factor, highlighting the persistent need to address both behavioral and structural determinants, including poverty and low education, in IPV prevention efforts. These findings carry significant implications for policy, practice, and the Ecological

Model of Violence Prevention. To effectively reduce IPV, programs should prioritize expanding safe and private digital access for women, and mandate the integration of gender-sensitive digital literacy and secure technology use training. Public messaging through diverse platforms should actively promote respectful relationships, positive masculinity, gender equality, focusing on impactful content. Simultaneously, community-level alcohol-reduction programs and poverty-alleviation initiatives, such as conditional cash transfers and job skills training, are crucial for addressing the underlying structural factors contributing to women's vulnerability to IPV. Future research should employ longitudinal designs and qualitative methods to elucidate further the dynamic interplay between digital access, partner dynamics, and women's safety in Cambodia.

## Supporting information

**S1 Table. Variance Inflation Factor (VIF) for Sexual Violence Model.**
(DOCX)

**S2 Table. Variance Inflation Factor (VIF) for Physical Violence Model.**
(DOCX)

**S3 Table. Variance Inflation Factor (VIF) for Emotional Violence Model.**
(DOCX)

**S4 Table. Variance Inflation Factor (VIF) for Any Intimate Partner Violence Model.**
(DOCX)

**S1 File. DHS Approval Letter for CDHS Data Access.**
(PDF)

## Acknowledgments

We thank the DHS program for giving permission to use the CDHS 2022 datasets.

## Author contributions

**Conceptualization:** Samnang Um, Sopheap Suong, Chantrea Sieng, Sovandara Heng, Grace Marie Ku, Sothy Heng.

**Data curation:** Samnang Um, Chantrea Sieng, Sovandara Heng, Grace Marie Ku, Sothy Heng.

**Formal analysis:** Samnang Um, Chantrea Sieng, Sovandara Heng, Grace Marie Ku, Sothy Heng.

**Funding acquisition:** Samnang Um.

**Investigation:** Samnang Um, Grace Marie Ku, Sothy Heng.

**Methodology:** Samnang Um, Sopheap Suong, Sovandara Heng, Grace Marie Ku, Sothy Heng.

**Project administration:** Samnang Um, Chantrea Sieng, Sovandara Heng, Sothy Heng.

**Resources:** Samnang Um.

**Software:** Samnang Um.

**Supervision:** Samnang Um, Grace Marie Ku, Sothy Heng.

**Validation:** Samnang Um, Grace Marie Ku, Sothy Heng.

**Visualization:** Samnang Um.

**Writing – original draft:** Samnang Um, Sopheap Suong, Chantrea Sieng, Sovandara Heng, Grace Marie Ku, Sothy Heng.

**Writing – review & editing:** Samnang Um, Sopheap Suong, Sovandara Heng, Grace Marie Ku, Sothy Heng.

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
