## [Decision Letter · Decision Letter 0]

30 Oct 2025

PLOS ONE

Dear Dr. Um,

After careful consideration, we are pleased to inform you that your manuscript has merit and shows potential for publication. However, it requires **major revision** before it can be considered for acceptance. We therefore invite you to submit a revised version that comprehensively addresses the reviewers’ comments and the additional points outlined below.

The presentation of terms "**violence types: physical, sexual, emotional, and intimate partner violence (IPV)** " leads to ambiguity because the text often lists IPV as if it were a distinct form of domestic violence alongside the three specific forms of violence. For instance, the abstract states the study examined associations with "**violence types: physical, sexual, emotional, and intimate partner violence (IPV)** ". Similarly, the outcome variable section defines the outcome as experience of domestic violence, specified as "**sexual violence, physical violence, emotional violence, and intimate partner violence (IPV)** ". This simultaneous listing of the specific types and the composite term (IPV) makes the categories appear mutually exclusive or confusingly similar. IPV, child abuse, in-law abuse, etc., are nuances or types of domestic violence; this should not be mixed with physical, emotional and sexual abuse. Therefore, while the authors need to adjust their phrasing in the text (e.g., in the abstract and outcome variable description) to prevent the confusion that IPV is a fourth, separate form of violence, the definition provided confirms that IPV is the overarching term representing the experience of *any* of the three specific violence forms.

The manuscript **does not explicitly name or detail a formal theoretical framework** (such as the Ecological Model or Empowerment Theory) in a dedicated section. Although the study discusses conceptual pathways related to autonomy and resource access, it does not formalize these concepts within a recognized theory.

The manuscript explicitly states that in the adjusted models, **motorcycle ownership and media exposure did not show statistically significant associations with IPV** . The discussion of these null findings attributes the lack of association to the possibility that "**mere access or exposure to these resources may not be sufficient to alter the underlying power dynamics that contribute to violence** ". While this is a conceptual interpretation, the authors do not extensively *debate* why motorcycle ownership, which theoretically enhances mobility and economic participation, might fail to demonstrate a protective effect in Cambodia, apart from stating that physical mobility alone may not be enough without shifts in gender norms

The manuscript's policy suggestions, while present, are highly concentrated in the abstract and conclusion sections, and are often phrased as broad programmatic needs rather than specific policy mechanisms. For example, the conclusion suggests that programmes should prioritise "**expanding safe and private digital access for women** " and integrating "**digital literacy and secure technology use training** ". While these are necessary steps for implementation (programmatic outcomes), they require underlying policy decisions (e.g., national strategy for secure digital inclusion, mandated funding for literacy programmes) that are not explicitly detailed.

We look forward to receiving your revised manuscript.

Kind regards,

Lanre Abdul-Rasheed Sulaiman, PhD

Academic Editor

PLOS ONE

Journal Requirements:

Reviewers' comments:

Reviewer's Responses to Questions

**Comments to the Author**

1. Is the manuscript technically sound, and do the data support the conclusions?

Reviewer #1: Yes

Reviewer #2: Yes

2. Has the statistical analysis been performed appropriately and rigorously?

Reviewer #1: Yes

Reviewer #2: Yes

3. Have the authors made all data underlying the findings in their manuscript fully available?

Reviewer #1: Yes

Reviewer #2: Yes

4. Is the manuscript presented in an intelligible fashion and written in standard English?

Reviewer #1: Yes

Reviewer #2: Yes

Reviewer #1: The authors made significant contribution to discourse on domestic violence. While the scholarship was cohesive and comprehensible, the authors did not clearly project policy implication of the study. I could deduce programmatic outcomes, but policies make these interventions more sustainable.

Reviewer #2: Reviewer’s comments

The article examined how digital access, media exposure, motorcycle ownership and partners' alcohol use are associated with violence types: physical, sexual, emotional, and intimate partner violence (IPV), while controlling for socio-demographic factors. This is timely and relevant perhaps with the rising cases of intimate partner violence in our contemporary society.

However, the authors need to clarify ‘violence types’ and ‘intimate partner violence (IPV)’. The way they are being currently used in the article are confusing or similar: violence types (physical, sexual and emotional) and intimate partner violence. There is the need for further clarifications.

Also, the authors have not provided clear and convincing evidence on the novelty of this study in Cambodia. I have seen a number of empirical studies on how digital access, media exposure and partners' alcohol affect intimate partner violence (IPV). The only novelty I could agree with is ‘motorcycle ownership’.

Also, the article lacks literature review where gaps ought to have emerged. Due to lack of literature, there are no conceptual, theoretical and empirical debates and engagements. Furthermore, the study does not have clear theoretical framework.

While methodology section contains important elements, it lacks justifications. The authors have not really justified most of their methodological choices. The article will benefit from proper and adequate justifications of each methodological choice made.

Generally, the analysis is good. Nonetheless, the authors need to improve their interpretations of key statistical results. In other words, the interpretations of key figures should be clear and explicit.

In addition, the authors need to improve on discussion of findings. For instance, novel findings should be incorporated with proper interpretations and justifications. This could be linked to dearth of conceptual, empirical and theoretical review. Also, implications to policies, practices and theories should be more explicit.

Overall, the article should be accepted for publication after the above issues have been attended to.

**Do you want your identity to be public for this peer review?** For information about this choice, including consent withdrawal, please see our Privacy Policy

Reviewer #1: No

Reviewer #2: **Yes:** Dr. Moshood Issah

---

## [Author Response · Author response to Decision Letter 1]

20 Dec 2025

Response to Reviewers

Manuscript ID: PONE-D-25-40112 Title: Determinants of Domestic Violence Against Women in Cambodia: How Digital Access, Media Exposure, Motorcycle Ownership, and Partners' Alcohol Use Matter

Journal: PLOS ONE

Academic Editor: Lanre Abdul-Rasheed Sulaiman, PhD

Dear Dr. Sulaiman and Reviewers,

Thank you for the insightful feedback on our manuscript. We appreciate your positive assessment of the study's merit and timely relevance. We have undertaken a major revision, addressing every point raised by the Academic Editor and the Reviewers. We believe these revisions significantly strengthen the manuscript's theoretical grounding, conceptual clarity, interpretation of findings, and policy relevance.

We have uploaded the revised files: 1) ‘Response to Reviewers’, 2) ‘Revised Manuscript with Track Changes,’ and 3) ‘Manuscript’ (clean version). We thank you again for the opportunity to strengthen our manuscript and look forward to hearing from you.

Sincerely,

Samnang Um, PhD

---

## [Decision Letter · Decision Letter 1]

4 Mar 2026

Determinants of Domestic Violence Against Women in Cambodia: How Digital Access, Media Exposure, Motorcycle Ownership, and Partners' Alcohol Use Matter

PONE-D-25-40112R1

Dear Dr. Um,

We’re pleased to inform you that your manuscript has been judged scientifically suitable for publication and will be formally accepted for publication once it meets all outstanding technical requirements.

Kind regards,

Russell Kabir, PhD

Academic Editor

PLOS One

Additional Editor Comments (optional):

Reviewers' comments:

Reviewer's Responses to Questions

**Comments to the Author**

Reviewer #1: All comments have been addressed

2. Is the manuscript technically sound, and do the data support the conclusions?

Reviewer #1: Yes

3. Has the statistical analysis been performed appropriately and rigorously?

Reviewer #1: Yes

4. Have the authors made all data underlying the findings in their manuscript fully available?

Reviewer #1: Yes

5. Is the manuscript presented in an intelligible fashion and written in standard English?

Reviewer #1: Yes

Reviewer #1: (No Response)

**Do you want your identity to be public for this peer review?** For information about this choice, including consent withdrawal, please see our Privacy Policy

Reviewer #1: No

---

## [Editor Report · Acceptance letter]

PONE-D-25-40112R1

PLOS One

Dear Dr. Um,

I'm pleased to inform you that your manuscript has been deemed suitable for publication in PLOS One. Congratulations! Your manuscript is now being handed over to our production team.

Kind regards,

on behalf of

Dr. Russell Kabir

Academic Editor

PLOS One